# The Role of Birds of Prey in West Nile Virus Epidemiology

**DOI:** 10.3390/vaccines8030550

**Published:** 2020-09-21

**Authors:** Beatriz Vidaña, Núria Busquets, Sebastian Napp, Elisa Pérez-Ramírez, Miguel Ángel Jiménez-Clavero, Nicholas Johnson

**Affiliations:** 1Bristol Veterinary School, University of Bristol, Bristol BS40 5DU, UK; 2IRTA, Animal Health Research Centre (CReSA IRTA-UAB), 08193 Bellaterra, Spain; nuria.busquets@irta.cat (N.B.); sebastian.napp@irta.cat (S.N.); 3Animal Health Research Centre INIA-CISA C, 28130 Madrid, Spain; elisaperezramirez@gmail.com (E.P.-R.); majimenez@inia.es (M.Á.J.-C.); 4Virology Department, Animal and Plant Health Agency, Addlestone KT15 3NB, UK; Nick.Johnson@apha.gov.uk

**Keywords:** West Nile virus, birds of prey, raptors, infection, epidemiology, diagnostic, *Flavivirus*, encephalitis, vaccine

## Abstract

Reported human cases of West Nile virus (WNV) in Europe increased dramatically in 2018. Lineage 1 strains had been circulating in Euro-Mediterranean countries since the early 1990s. The subsequent introduction of WNV lineage 2 has been responsible for the remarkable upsurge of European WNV outbreaks since 2004, including the dramatic increase in human cases observed since 2018. The virus exists in a natural cycle between mosquitoes and wild birds, with humans and horses acting as dead-end hosts. As the key vertebrate hosts in the transmission cycle of WNV, avian species have been the focus of surveillance across many countries. Raptors appear particularly susceptible to WNV infection, resulting in higher prevalence, and in some cases exhibiting neurological signs that lead to the death of the animal. In addition, birds of prey are known to play an important role as WNV reservoir and potentially amplifying hosts of infection. Importantly, raptor higher susceptibility/prevalence may indicate infection through predation of infected prey. Consequently, they are considered important target species when designing cost-effective surveillance for monitoring both seasonal WNV circulation in endemic countries and its emergence into new areas, where migrating raptors may play a critical role in virus introduction. This review summarizes the different aspects of the current knowledge of WNV infection in birds of prey and evaluates their role in the evolution of the epizootic that is spreading throughout Europe.

## 1. Introduction

West Nile virus (WNV) is a zoonotic agent that is maintained in a transmission cycle between birds and mosquitoes. WNV is member of the genus *Flavivirus* of the family *Flaviviridae*. WNV strains are classified into at least 7 genetic lineages [1,2] with highly pathogenic isolates mainly belonging to lineage 1 (L1) and 2 (L2) strains [3]. It causes a febrile disease in humans and horses that in some cases progresses to fatal encephalitis. Even though several human vaccine candidates have been developed for WNV, none has been licensed and currently there are no available WNV vaccines for humans [4].

WNV L1 was first isolated in Africa [5,6] from where it spread to the Middle East and Europe during the mid-twentieth century, and finally to the United States (US) in 1999 [7,8,9]. Since 2004, a new WNV L2 is circulating in Europe [1,10] and is responsible for the unprecedented increase of WNV human cases registered since 2018 [11,12,13]. In Europe, outbreaks in human and equine populations have usually been documented in Mediterranean countries. However, new WNV cases have recently been reported in humans, avian, and equine species in Germany at higher latitudes than in previous years [14]. 

As the key vertebrate hosts in WNV transmission cycle, avian species are the focus of surveillance across the world [15,16,17]. Birds of prey are target species in these surveillance programs because some species are very susceptible to infection and exhibit a wide range of clinical signs. In fact, WNV infection is the most frequently diagnosed infectious disease among raptors in the US and Canada [17,18,19]. 

Birds of prey or raptors, derived from the latin verb *rapio* (to seize), are those predatory birds that catch prey using sharp talons, or claws, for grasping and killing prey, and a powerful curved beak for tearing flesh. The group is not taxonomically valid but includes members from the orders Accipitriformes (eagles, ospreys, kites, true hawks, buzzards, harriers, and vultures), Falconiformes (falcons and caracaras) and Strigiformes (owls). This large group of birds are found throughout the world. WNV is transmitted to raptors mainly through mosquito bites. However, oral transmission through the consumption of infected prey or carrion has also been described [20]. 

One of the main challenges when dealing with diseases of wildlife is the limited access to the animals of interest. When wildlife becomes infected, determining the clinical response and outcomes are difficult to establish when such events are seldom observed. In the case of birds of prey and WNV, some of these questions have been addressed using (i) observations in captive birds usually from wildlife rehabilitation centers (WRC), (ii) active surveillance programs, and (iii) experimental infections. 

The possible role of the birds of prey as reservoirs, spreaders, or sentinels of WNV is not clear. Given that those terms are loosely defined, for the purpose of this review, we used the following definitions: (a) reservoir for a species that is likely to get infected, be infectious, and resident of a particular geographical area (i.e., non-migratory); (b) spreader for a species likely to get infected, be infectious, but migratory; and (c) sentinel for a species also likely to get infected, be non-infectious or present a low infectious risk, and resident.

The following sections review the biological, epidemiological, and clinical aspects of WNV infection in birds of prey. It ends up with a conclusion reviewing this data against the different epidemiological roles of raptors as reservoirs, sentinels, or spreaders of WNV infection. 

## 2. Infection and Immunity

### 2.1. Infection in Birds of Prey

Observing clinical signs in wildlife is challenging, given that wild birds are often difficult to observe and sparsely distributed. Detecting disease is often easier in domestic animals, but in the case of WNV most poultry species, such as the chicken (*Gallus gallus*), appear refractory to overt clinical disease. The exception is the domestic goose (*Anser anser domesticus*), where WNV-associated mortality has been previously reported [21,22,23].

For birds of prey, the pathogen is usually detected in the carcasses of birds found dead or moribund through wildlife surveillance programs, raptors admitted to WRC [24], or following mass mortality events [20]. One raptor species in Europe that has repeatedly been associated with WNV infection is the Northern goshawk (*Accipiter gentilis*) [17,18,25,26,27,28]. The reason for this is unclear, although the increased incidence observed in Northern goshawks (*Accipiter gentilis*) may be associated to the species predation on smaller birds that can also act as a reservoir for WNV, as demonstrated by experimental raptor infection through feeding of WNV-infected prey [29].

In most cases, evidence for WNV-associated disease is obtained in wild dead birds on pathological investigation. In addition, confirmation of WNV infection may occur following the observation of clinical disease in captive birds. For example, in Canada, shortly after the introduction of WNV in the country, a large number of captive North American owls (108 out of 245) died at a WRC in Ontario [30]. Necropsy samples of brain, lung, liver, and spleen from 85 birds were tested and 79 were confirmed WNV positive by RT-PCR. Again, captive raptors, both those born and raised in captivity, and those that are being rehabilitated, can form a cornerstone of a WNV surveillance program due to their susceptibility to infection [17]. 

A small number of experimental infections have been attempted on raptor species and have reported mixed findings with respect to clinical disease and infection outcomes. A large study of five different raptor species including the American kestrel (*Falco sparverus*), Golden eagle (*Aquila chrysaetos*), Red-tailed hawk (*Buteo jamaicensis*), Barn owls (*Typto alba*) and Great-horned owls (*Bubo virginianus*) assessed three methods of WNV infection routes: oral infection with infected mice; infected mosquitoes; and direct needle inoculation with an L1 strain. However, no clinical signs were observed in this study or in experimentally infected Gyrfalcons (*Falco rusticolus*) [31,32]. In contrast, experimental infection of large falcons resulted in neurological disease and death after subcutaneous needle inoculation with higher challenge doses of an L1 strain and after inoculation with different doses of an L2 strain [33]. Another factor that may influence the outcome of WNV infection in raptors is the particular species-specific susceptibility. In general, owls appear to be more likely to develop neurological clinical signs than other raptors, and among owls, those from northern species, such as the Great Horned owl (*Bubo virginianus*) and Snowy owl (*Bubo scandiacus*), seem more susceptible [30]. However, statistically significant differences in mortality rates and WNV prevalence have not been found among taxonomic orders, age class, or sex, with the exception of immature Red-tailed hawks (*Buteo jamaicensis*), which were found to be more susceptible than adult Red-tailed hawks [34].

### 2.2. Immunity against WNV in Birds of Prey

In birds, as in mammals, protection against WNV is determined by the presence of antibodies in the blood of the individual. This can be measured by a range of serological assays, the most stringent being a virus neutralization test (VNT), which detects serum neutralizing antibodies and more accurately detects protective antibodies. The main alternative is the capture enzyme-linked immunoassay (ELISA) that detects antibodies directed against the virus envelope protein. Also, there are class-specific ELISAs for the detection of WNV Immunoglobulin (Ig) Y (the avian equivalent of IgG in mammals) or IgM. In adult birds, antibodies are developed following exposure to, and infection with WNV. The detection of such antibodies in an apparently healthy bird suggests either past asymptomatic infection or recovery from a non-fatal infection [35]. Seropositive birds are likely protected against infection in future exposure to WNV, as assayed in experimental studies in some bird species such as the House sparrow (*Passer domesticus*) [36]. However, no experimental data on long-term humoral immunity duration is available for raptors.

Longitudinal sampling of raptors held in WRC has confirmed the seroconversion of birds of prey in response to the seasonal emergence of WNV in North America [37]. Different studies have detected seropositive birds of prey in both Europe [38,39] and the Americas [40,41]. These findings suggest past WNV infection but recovery in otherwise healthy birds. Experimental infections in raptors have consistently shown the induction of anti-WNV antibodies, both total and neutralizing antibodies, from 6 dpi [31,32,33]. The exception to this is the presence of maternal antibodies in juvenile birds. IgY is present in the egg yolk and crosses the yolk sac membrane to enter the bloodstream of the developing embryo. If the mother has developed anti-WNV antibodies, these will be transferred conferring temporary immunity to the young. In raptors, maternal transfer of WNV neutralizing antibodies was demonstrated in a captive breeding colony of Eastern screech-owls (*Megaschops asio*) [42]. However, subsequent studies in the House sparrow suggest that maternally-derived antibodies decline rapidly, being undetectable after 9 days, and do not confer protection to the juvenile after this point [43]. Maternal antibodies were suggested as a potential reason for the absence of disease in Red-footed falcon (*Falco vespertinus*) nestlings in the presence of infected mosquito vectors [44]. Avian immunity research has focused on humoral immunity, i.e., the induction of antibodies. However, there is a dearth of information on the cell-mediated responses in avian species and how these assists in controlling WNV infection.

## 3. WNV Transmission 

Different studies carried out in both Europe and North America indicate that raptors are among the birds most frequently infected during WNV outbreaks [45,46], although the reason why that occurs is unknown. In raptors, as in any other birds, three mechanisms of WNV infection may be considered: mosquito-borne transmission, contact transmission, and oral transmission.

### 3.1. Mosquito-Borne Transmission

The most frequent mechanism of WNV infection in birds is by mosquitoes that feed on a viremic animal (most likely another bird) [47]. The lower threshold of serum viremia considered infectious to mosquitoes from birds ranges around 5 log10 PFU/mL [29,33,48,49]. *Culex* spp. mosquitoes are considered the main vectors, but some species from other genera, for example *Aedes albopictus,* are also competent for WNV transmission [50]. In addition to mosquitoes, WNV has been isolated from other hematophagous ectoparasites such as ticks, although their role in WNV transmission is not clear [51].

In an experiment to evaluate the feeding preference of *Cx. pipiens*, the main vector in Europe and one of the most important in North America, using 8 bird species from 5 different orders (Passeriformes, Strigiformes, Columbiformes, Falconiformes and Anseriformes), Llopis *and colleagues* found a clear preference for large raptors [52]. A similar preference of mosquitoes for certain species of raptors (e.g., Great horned owl) was found in a study carried out in a WRC in Alabama (US) [53]. Mosquito feeding preferences may be influenced by several factors such as defensive behavior of the host, body heat, the production of carbon dioxide, or size [54]. The feeding success of *Cx. nigripalpus* was found to be highest in nocturnal raptors (Strigiformes) compared to several other avian species, and a possible explanation was their weaker protective behavior. Interestingly, while in most birds the highest body temperature is reached during the day, in most species of owls this peak occurs at night, coinciding with the maximum mosquito activity [55]. Mosquitos are also attracted by carbon dioxide, and larger animals exhale higher quantities of carbon dioxide [52]. Therefore, raptors, many species of which may be considered as large compared with other birds, would be expected to be more attractive to mosquitoes.

### 3.2. Contact Transmission

Another mechanism of WNV transmission is through direct contact with infected birds as some bird species may shed large amounts of virus in oral and cloacal secretions, which may result in the infection of contact birds through the oral route [29,47,56]. This type of transmission has been demonstrated in birds of the Corvidae and Laridae families, while contact transmission could not be proven when assayed experimentally in three raptor species. Contact transmission may be epidemiologically relevant when large numbers of birds concentrate in the same area (in breeding colonies or stopovers during migration), so in the case of raptors, which do not tend to aggregate, this may not be an important form of transmission when compared to predation.

### 3.3. Oral Transmission

WNV can also be transmitted by feeding on infected birds and other vertebrate hosts, such as reptiles, amphibians and mammals which are also susceptible to WNV and serve raptors and other birds either as prey or carrion [57,58,59,60]. Several species of birds (e.g., Great horned owl or American crow (*Corvus brachyrhynchos*)) have experimentally proven their susceptibility via this mechanism [29]. Long-term persistence of WNV in tissues of infected animals would increase the probability of infection of predator birds. This may result in the infection of birds by prey ingestion, even months after the end of the mosquito season, providing a mechanism of overwintering. This was the likely explanation for the detection of WNV in the brain of a Red-tailed hawk in New York in February, a period with no mosquito activity [61]. Further evidence of the role of WNV oral transmission in raptors was found in Utah (US) in December 2013, after more than 40 Bald eagles (*Haliaeetus leucocephalus*) were found dead, not long after a mortality event which involved thousands of Eared grebes (*Podiceps nigricollis*) [20]. Given the timing of the outbreak, mosquito-borne transmission was unlikely, and, because lakes and ponds were ice-covered, this prevented bald eagles from fishing, their main source of food. Therefore, WNV exposure by scavenging on infected grebes’ carcasses (ten out of ten Eared grebes tested by RT-PCR were WNV positive), seemed the most likely explanation. In Europe, WNV infection has been repeatedly detected in Northern goshawks [46], and interestingly, goshawks feed mainly on birds [62], which if infected may cause the infection of the predator. In fact, detection of WNV in Northern goshawks occurred in the same area where a high prevalence of infection was repeatedly detected in Eurasian magpies (*Pica pica*) in Spain [63,64]. 

The importance of oral transmission is likely to vary among the species of raptors depending on their diet, the susceptibility to WNV of the animals they prey on, as well as the persistence of the virus in tissues of those animals. Some raptors feed preferentially on birds, but depending on the species they eat, their risk of WNV infection will vary significantly. Other raptors may essentially prey on mammals or reptiles, which may also be infected by WNV [65]. However, further research is needed about WNV prevalence and persistence in different species of mammals and reptiles.

## 4. Clinical Features and Gross Pathology

Clinical signs reported in WNV-infected raptors vary widely between species and among individuals. The clinical outcome can range from asymptomatic or unspecific clinical signs to severe neurological signs and sudden death. The development of clinical signs in birds of prey is caused by viral invasion of the central nervous system (CNS) and/or other organs such as the heart, liver, spleen, and kidney [66]. After experimental infection, clinical signs may be absent or appear approximately 5–8 dpi in hawks and owls, and neurological signs may appear from 8–9 dpi, increasing in severity with the course of infection [29,31,32,49]. Interestingly, many WNV-positive raptors often show ocular lesions and evidence of trauma [34,67], as well as concurrent lesions of other non-viral infectious diseases, such as bacterial septicemia, aspergillosis and *Leucozytozoon* spp. infection [45,63,68,69]. Therefore, it is important to consider concurrent WNV infection especially in birds with traumatic injuries, where WNV might have contributed to death. 

### 4.1. Non-Specific Clinical Signs 

Various non-specific clinical signs associated with WNV infection have been observed among raptor species. Most commonly described signs include dehydration, emaciation, depression, weight loss, lethargic lying on the ground often with extended wings, crouched body postures with drooped wings, recumbency and sudden death [63,68]. In addition, feather abnormalities (pinched-off mature or bloody feathers, poor feather condition and fluffed feathers), dysphagia, decreased vocalization, hyperthermia, and greenish discolored uric acid in excrements have also been described (Table 1). Non-specific clinical signs were frequently the only clinical sign observed in Golden eagles, Eastern screech owls (*Megascops asio*), Great grey owls (*Strix nebulosa*), Barn owls and American kestrels. Peregrine falcons (*Falco peregrinus*) and Swainson’s hawks (*Buteo swainsoni*) frequently presented no clinical signs apart from sudden death [18,67,70].

### 4.2. Neurologic and Ocular Clinical Signs

Neurological signs in WNV-infected birds of prey also differ widely between species and individuals. Clinical signs vary in severity as well as in time course with some Great horned owls and Golden eagles presenting continuous or intermittent neurological signs for years after infection [18]. Neurological signs commonly reported include tremor and head shacking “bobble head”, convulsions of legs and wings, seizures, ataxia, torticollis, head tilt, incoordination (reduced ability to perch and stand), disorientation, hind limb paresis or rigidity, dullness, circling and coma (Table 1). Among raptor species, Bald eagles, Northern owls, Northern goshawks, and Ferruginous hawks (*Buteo regalis*) commonly develop neurological signs after WNV infection [67,68,70,71]. 

Ophthalmologic signs are frequently observed in Bald eagles, Cooper’s hawks (*Accipiter cooperii*) and Great horned owls [18,68,70,72] and mostly consist of blindness, visual impairment, nystagmus, and abnormal pupillary responses (Table 1). 

### 4.3. Gross Pathological Lesions

Post-mortem macroscopic lesions observed in birds of prey are summarized in Table 1. Gross pathological lesions are more commonly observed in hawks than in owls [73]. However, gross lesions are infrequent in all raptor species in comparison to clinical signs or histopathological lesions [25,31,45,74]. The most frequently described macroscopic lesion among WNV-infected birds of prey is emaciation or cachexia characterized by a reduced or a lack of subcutaneous and organ fat deposits and atrophy of pectoral muscles. Other frequently reported gross lesions include calvarial and meningeal hemorrhages. However, those may be related to concurrent trauma. In addition, mild to moderate hepato- and/or splenomegaly, and multifocal discoloration of the myocardium with/without subepicardial petechiae and ecchymoses are frequently observed in some hawks and owls. Occasionally, mottled kidneys, serous atrophy of bone marrow, and gastrointestinal dilation and hemorrhages have been described (Table 1). Finally, cerebral atrophy and malacia have been described in Great horned owls, Red-tailed hawks and Northern goshawks, presenting with more severe and chronic neurological signs [31,45,70,74]. 

## 5. Histopathology and WNV Antigen Distribution

Differences in WNV pathology in raptors likely arise from a combination of factors related to the host and the virus strain involved. Host factors are linked to genetic variation between species, but also within species in different populations, and even at the individual level [3,29,86]. On the other hand, virus factors, such as the presence of virulence determinants in the viral genome, also influence the outcome of the infection in each host [3,87]. Non-progressive, acute, or more prolonged course of disease will also partly affect the severity and distribution of lesions and viral antigen detection in different organs. Lesions are highly variable, both within and between species, ranging from focal and/or very mild to severe and diffuse with raptors presenting clinical signs usually having more severe lesions [88]. In general, owls and falcons with more chronic courses of disease show more severe histological lesions [31,32,33,49].

Histopathologic lesions in raptors are more commonly observed in the heart and brain, while the heart and kidney were the organs more commonly reported to show WNV positivity by immunohistochemistry (IHC) [45]. Differences in pathological lesions caused by WNV L1 and L2 have been observed in experimentally infected Gyrfalcons. However, these are probably related to differences in the time course of the disease induced by each lineage, with longer clinical courses usually showing more severe histopathological changes [33]. Splenic lymphoid depletion, nephritis and hepatitis were present in L2 infected goshawks while L1 infected birds presented myocardial necrosis [70,79,80]. 

Due to the limited observations directly comparing lineages, it is not possible to report a characteristic pattern associated with a particular lineage. Susceptibility to WNV infection varies among raptor species, which is also reflected by differences in the location and severity of lesions. WNV positive staining showed high variability in distribution between species (Table 1). Amongst hawks, lesions seemed to be more severe in Red-tailed hawks and Sharp-shinned hawks (*Accipiter striatus*), while Cooper’s hawks presented with milder lesions [73]. Curiously, WNV immunolabelling in the CNS of owls was found to be inconsistent even in the presence of typical histological lesions and antigen positivity in other organs, which usually was widespread in most owl species [74]. 

### 5.1. Heart, Skeletal Muscle and Kidney

Myocardial lesions are the most common and most severe lesions reported in raptors, particularly in hawks [72,73,74]. Lesions are characterized by a necrotizing lymphoplasmacytic and histiocytic myocarditis (Figure 1).

Occasionally, fine mineralization and fibrosis in the myocardium and epicardium, and myocardial and epicardial hemorrhages, are also described [74]. Lesions in skeletal muscle (as described per myocardium) have been mainly described in pectoral muscles with varying degrees of severity. Positive WNV immunolabelling has been reported in myocardiocytes and myofibers with or without concomitant histological lesions [33,66]. A lymphoplasmacytic interstitial nephritis with interstitial infiltrates and moderate necrosis of the tubular epithelium has been described in kidneys. Lesions were associated with WNV antigen positivity in tubular and collecting epithelial cells, as well as in interstitial fibroblasts [66,73]. 

### 5.2. CNS and Eye

CNS lesions in WNV-infected raptors are consistent with meningoencephalitis and myelitis characterized by neuronal necrosis and degeneration with varying perivascular and meningeal lymphoplasmacytic infiltrations and glial nodules (Figure 2). Less commonly, some individuals also present with sciatic neuritis and/or sciatic nerve WNV antigen positivity (Table 1). CNS lesions are commonly found in the cerebellum (mostly in the molecular layer), brainstem, and cerebrum with the meninges only slightly affected if at all [70,73,77,88]. Lesion location varies between species. For example, hawks generally show more prominent lesions in the cerebrum, while cerebellar lesions are more common in owls, with the exception of Snowy owls [72,73,80,81]. As a result of cerebellar lesions, northern owls frequently present head tremors, but, in contrast, they less often show signs of impaired vision, which are frequent in hawks. In accordance with these observations, higher numbers of WNV positive cerebral neurons are detected in hawks [77] than in cerebellar Purkinje cells, while higher numbers of WNV positive cerebellar Purkinje cells are observed in owls and in the Spanish imperial eagle (*Aquila adalberti*) than in cerebral neurons [69,73,88]. 

Optic neuritis, iridocyclitis, and pectinitis are mostly described in hawks and eagles (Table 1). Retinal lesions are classified as type I and characterized by lymphoplasmacytic infiltration in the subjacent choroid with degeneration limited to the outer retina, which is associated with acute ophthalmic disease and viral antigen presence in the retina [74,78,83].

### 5.3. Spleen, Liver and Lymphoid Organs

Splenic lesions in WNV-infected raptors are in general more subtle than in other organs and consist of lymphoid depletion with multiple small foci of necrotic/apoptotic lymphocytes and necrosis of splenic ellipsoids [74,81]. WNV antigen detection has been reported in macrophages, Kupffer cells, and blood monocytes of the liver and spleen [66,73,77]. In addition, WNV positivity by IHC has been described in the thymus and bursa of Fabricius in some birds [74]. A lymphoplasmacytic and histiocytic hepatitis has been commonly reported in several raptor species, accompanied by biliary duct hyperplasia in Bald eagles (Table 1).

### 5.4. Lungs and Other Organs

Lung lesions described as a mild increase in the number of lymphocytes and plasma cells around bronchioles have been infrequently reported in infected birds of prey. In the lungs, WNV antigen has been detected in epithelial lung cells and macrophages in several species, although not always associated with lesions [66,73,74]. Rarely, mild lymphoplasmacytic lesions have been found in the pancreas, thyroid, skin, trachea, and various gastrointestinal organs including oropharynx, esophagus, ventriculus, proventriculus and intestine, sometimes accompanied by concomitant WNV antigen positivity in the epithelial and follicular cells of affected organs. In a few cases, steroid producing cells of various endocrine organs, and oocytes and stromal cells in the ovaries showed WNV positivity by IHC, without associated histological lesions (Table 1).

## 6. Experimental WNV Infection

Despite the numerous reports of WNV natural infections in birds of prey, precise information of the pathogenesis of the disease derived from experimental studies is very limited. Out of the 557 species of raptors that exist in the world, only seven species (1.2%) have been experimentally infected with WNV (Table 2). This reduced number, as compared with other taxonomic groups of birds [47], is probably related to the difficulties to obtain the birds, ethical considerations on using endangered species, and to properly maintain and handle these species in Biosafety Level-3 (BSL-3) facilities. 

The first experimental WNV studies with birds of prey were performed in Common kestrels (*Falco tinnunculus*). The studies were conducted by Work and colleagues in 1955 using the L1 Egyptian strain Ar-248 [89] and no more experimental trials were done with raptors until the virus reached for the first time the American continent in 1999. After that, several studies were carried out in the US using the prototype strain NY99 [29,31,49]. In fact, our knowledge on the effect of WNV infections in raptors is highly biased towards this L1 strain (Table 2). Despite the increasing incidence of L2 in Central and Eastern Europe, only one study has so far investigated the disease caused by an L2 strain (Austria 2009) in falcons [33]. More experimental trials with other European or African L1 and L2 strains are needed to gather unbiased knowledge of the impact of WNV infections on raptor populations outside the U.S.

The experimental trials performed until now offer limited but very useful information about the pathogenesis, host response, and competence capacity of raptors after WNV infection. These studies have demonstrated that the seven assayed species are susceptible to the infection and that most of them suffer a systemic infection as the virus can be detected in several organs up to 2–3 weeks after subcutaneous inoculation, mosquito bite, or oral exposure. The oral transmission of the virus through consumption of infected prey (mice) has been tested in 4 raptor species, and although in all cases the infection has been confirmed in at least one animal of the group, this route of infection seems much less efficient than through a mosquito bite [29,31,49]. In animals infected through subcutaneous inoculation, the virus is shed by the oral and fecal routes, although with lower viral loads in the latter. The presence of the infectious virus in feather pulp has been demonstrated in hybrid falcons and Eastern screech owls several days after infection [32,49]. This could facilitate the transmission of the virus by direct contact (i.e., feather picking) as observed in several avian species [47]. However, direct transmission of the virus could not be proved in any of the 3 raptors species where this infection route was tested [31,49] (Table 2).

The great variability in terms of morbidity and mortality has been observed after experimental infection. In most cases, the birds did not show any clinical signs [31,32,89], with the exception of Eastern screech owls and Gyr falcons that did suffer mortality (33–40%) following subcutaneous inoculation, particularly when challenged with an L2 strain [33] (Table 2). However, subclinical pathology was evidenced in infected birds in all the experimental trials and this could potentially affect survival in the wild much more than observed under controlled conditions with *ad libitum* food and water [31,32,33].

Likewise, striking variations in viremia levels and consequently in host competence capacity have been observed among raptor species. Of the assayed species, 5 developed high viremia levels (above the threshold 10^5^ PFU/mL necessary to infect a feeding mosquito) [29,33,49] and are therefore considered as competent hosts. By contrast, 2 other species and one hybrid were classified as not competent hosts since the infection elicited low viremia levels [31,32,89] (Table 2).

Only two studies have clearly provided comparable results to evaluate differences in pathogenicity between needle subcutaneous, mosquito bite, and oral transmission. In these studies, no overt clinical signs were observed, which may reflect that there is no difference between the infection/transmission route regarding the clinical outcome. Nonetheless, the number of infected animals was low in all studies N = 2–3 and therefore it is not possible to draw meaningful conclusions. On the other hand, viremic levels were higher in birds infected via mosquito bite, and the experimental studies where a few raptors presented clinical disease were inoculated subcutaneously, which may suggest that subcutaneous inoculation, and by extrapolating mosquito bite transmission, might be more pathogenic to susceptible raptors [29,31,49].

To accurately interpret the results derived from experimental infections in birds of prey, we must consider several limitations of this type of studies. First, the difficulties of obtaining the animals leads to low group sizes being used (always lower than 7 birds by the experimental group). Secondly, in most cases, the animals are non-releasable birds from WRC [29,31] or originate from captive breeding colonies [32,33,49], which implies that the birds may have pre-existing health conditions (including immunosuppression) that might affect the infection outcome. Finally, in the vast majority of the studies, the exact age or even the sex of the animals is not known or reported. 

Despite these limitations, experimental infection studies have been essential to elucidating the pathogenesis of the disease, identifying the main transmission routes, and determining host competence capacity in birds of prey.

## 7. WNV Epidemiology

Even though WNV infections have been reported from numerous species of birds of prey in Europe and North America, true rates of morbidity and mortality associated with WNV in raptor species in the wild remain unknown.

The probability that a WNV case in a raptor of a certain species is reported will be dependent on multiple factors, which can mainly be grouped to those related to the probability of infection, and those related to the probability of detection. Probability of infection depends on factors such as the feeding preferences of the main vector species, the animal species they eat (mammals, birds or reptiles and their WNV susceptibility), as well as the intrinsic susceptibility of the raptor species. The probability of detection in a given species will be conditioned by for example the intensity of clinical signs they develop after infection, the location of the infected raptor (e.g., proximity to a WRC), and obviously by the size of the population of that species (the larger the population, the more likely that infection will be detected). In general, raptor species with more explicit lesions and clinical symptoms are more likely to be detected.

### 7.1. North America

In the US between 1999 and 2004, 36 different raptor species were found dead with WNV infection by the CDC Arbonet Surveillance System [31]. The species most frequently found were Red-tailed hawks with 299 individuals, Great horned owls with 258, Cooper’s hawks with 145, Sharp-shinned hawks with 104, and American kestrel with 100. Those have also been the species most referenced in the reviewed epidemiological studies carried out in North America, with the addition of the Bald eagle (Table 3). Frequency of WNV reporting for some of those species may be highly influenced by their large populations in North America, for example, Red-tailed hawks (2,804,389 individuals), Great horned owls (3,788,535) or American kestrels (2,827,871), in contrast to the populations of Cooper’s hawks (845,663) or Sharp-shinned hawks (406,346) [90].

Frequency of reporting in Cooper’s hawks may be related to the fact that they mainly eat birds, including American robins and several kinds of jays [91], which are known to be key species for WNV transmission in North America [92]. Equally, ninety percent of the Sharp-shinned hawks’ diet is made of songbirds (i.e., Passeriformes), which are also highly susceptible to WNV infection [29,91]. In contrast, Red-tailed hawks eat mainly small mammals, American kestrels eat mostly insects and other invertebrates, while Great horned owls have a very diverse diet, which includes both mammals and birds [91,92].

Susceptibility to WNV infection varies among raptor species, which is reflected by differences in the location and severity of lesions, as well as the clinical symptoms developed, as described above. Within owls, Barn owls (family *Tytonidae*) appeared relatively resistant to WNV disease, and few cases were reported [18,73] despite their abundance in North America (3,460,224 individuals) [90]. Great horned owls and Barred owls (*Strix varia*) were found to be extremely tolerant to mosquitoes, with 90% and 82%, respectively, of recovered mosquitoes being fully blood-fed [93]. However, Great horned owls seem much more susceptible than Barred owls given the available studies (Table 3). 

### 7.2. Europe

In Europe, it is surprising how consistently WNV, in particular L2, has been detected from Northern goshawks (Table 4), particularly, since its population in Europe is small compared with other raptor species, which are much less frequently infected with WNV. There are only 166,000–220,000 pairs of Northern goshawks, compared to 403,000–582,000 of Eurasian Sparrowhawks (*Accipiter nisus*), 814,000–1,390,000 of Eurasian buzzards (*Buteo buteo*) or 409,000–603,000 of Common kestrels [94]. A clear example of the differences in susceptibility are the outbreaks in Central Europe in 2008–2009, where 45 positive Northern goshawks were detected as compared to one Sparrowhawk, and three Gyrfalcons (population 1100–1900) [28,46]. As with North American raptors, a possible explanation is diet. Northern goshawks feed mainly on birds, including a significant proportion of corvids [62,95], which are known to be highly susceptible for WNV infection [29]. In contrast, for Eurasian sparrowhawks, small birds make up the majority of their diet, while Eurasian buzzards and Common kestrels feed mainly on small mammals [95]. Besides Northern goshawks, WNV cases in raptors in Europe appear to be quite evenly distributed among different species (Table 4). A cause of concern is that several affected species, such as the Bearded vulture (*Gypaetus barbatus*), the Spanish imperial eagle, or the Snowy owl are considered as vulnerable according to the International Union for Conservation of Nature [96], which raises concerns regarding whether WNV may compromise the conservation of those species.

## 8. Diagnosis and Surveillance

West Nile virus diagnosis of birds of prey is very useful for WNV surveillance since many species are significantly affected by WNV infection, which may even cause the death of the animals. In fact, WNV circulation can be demonstrated in birds of prey since many species have tested positive for WNV both in North America (Table 3) [31,61] and in Europe (Table 4) [25,63,97]. Thus, early WNV detection in raptor clinical cases may trigger surveillance in other animal species, maximizing the possibility of WNV detection, which may be useful when the virus is circulating at low levels. Additionally, WNV surveillance in dead birds of prey could be adequate even after the end of the period of mosquito activity, as demonstrated with Red-tailed hawks, where the virus could be detected in winter, most probably transmitted via predation [61]. 

A variety of tissues including brain, heart, or liver can be used with success for viral detection in infected birds [66]. In those tissues, viral genome and antigens can be detected by RT-PCR and IHC, respectively [105]. Besides infected tissues, choanal, oral and cloacal swabs may be used to detect the virus [31,32,68]. It is worth mentioning that oral swabs were more sensitive to detecting viral shedding than cloacal swabs in some experimentally infected raptor species [49]. The viral detection via molecular techniques, such as RT-PCR, allows subsequent viral sequencing and phylogenetic analysis, which may enable the identification of viral strains circulating in specific areas. Additionally, serum samples from resident raptors can also be used to provide information of virus circulation since seroconversion or a significant increase in antibody titers in two serially collected specimen indicates recent WNV infection, as reported in Germany in 2018 [39]. 

Actually, clinic-admitted raptors diagnosed by both molecular and serological testing allowed the detection of virus circulation before other surveillance systems [106]. Serological tests such as ELISA allow rapid analysis of high numbers of samples and may allow detection of recent infection. However, it is important to bear in mind that, in contrast to molecular techniques, serological tests have several important drawbacks since they require (i) expertise in blood sampling; (ii) two samples for evidence of seroconversion or increasing antibody titers; and (iii) serum VNT, which are difficult to interpret because of cross-reactions with related flaviviruses, and moreover, they need to be performed at BSL-3 lab facilities. Surveillance based on molecular WNV diagnosis using swabs of raptors with general signs of illness, such as dehydration, emaciation, and debilitation with or without neurological signs, may be more feasible since samples can be easily provided by WRCs. This kind of surveillance may provide a reliable ante-mortem diagnosis of current WNV infection [68,107]. Additionally, bird samples (swabs and tissues) for WNV detection could be collected and shipped using FTA™ and RNASound™ cards [108], which preserve the viral genome and are easier to transport. However, it is important to consider that the virus will be inactivated by these cards, and therefore, it will not allow the isolation of the virus. 

WRCs, which receive high numbers of raptors, can certainly obtain samples for WNV diagnosis from birds in a cost-effective manner in comparison with other types of surveillance such as virus testing in the mosquito population during periods of low viral transmission [59]. Moreover, long-term (20 years) studies carried out in WRC in North America have revealed important data for WNV epidemiology such as: (i) the mortality attributed to infectious diseases, for which WNV was the most common etiology, and the most common cause of death after trauma and emaciation, (ii) WNV diagnosis in raptors during summer and fall reflected WNV seasonal activity [34] and, (iii) a syndromic surveillance suggested that monitoring of hawks showing WNV clinical signs could serve as a very good indicator of WNV circulation [17].

In this sense, it is worth highlighting Northern goshawks as an indicator of WNV activity [106] and ongoing emergence [28,81], since this species has been repeatedly found dead due to WNV infection in Europe [59,63,79,81,97]. Therefore, birds of prey from WRC should be included in all WNV surveillance programs since they can reflect the WNV-infection status of the area. 

## 9. Vaccination against West Nile Virus in Avian Species

Both mammals and avian species are susceptible to infection with WNV. However, infection in mammals leads to the development of viremia that is insufficient to transmit the virus to mosquitoes, and whilst susceptible to disease in the case of humans and horses, they are considered dead-end hosts. By contrast, avian species develop a much higher viremia, and act as the main vertebrate reservoir host that enables WNV to persist and spread. Currently, the only licensed vaccines in use against WNV are those for equids [109]. These have been in use for over ten years and are highly effective at preventing infection if used appropriately, specifically annual booster injections used to maintain protection. Despite the clear evidence of disease in some avian species, there are no licensed vaccines for use in birds, as recently reviewed by Jiménez de Oya *and colleagues* [110]. However, most vaccine candidates have demonstrated the development of anti-WNV antibodies in several bird species including raptors and some studies were able to confirm that these vaccines conferred partial protection from virus challenge [111,112].

Vaccination of falcons with Duvaxyn^®^ inactivated vaccine and Recombitek^®^ Equine WNV formulation resulted in relative protection in falcons in comparison to non-vaccinated animals [85]. In addition, DNA vaccines have been shown to provide protection against WNV in experimentally challenged large falcons [113]. Many zoos and wildlife centers in the US use licensed WNV horse vaccines in birds. However, the vaccine has not been tested for use in birds, and therefore, the safety and efficacy of its use in birds is neither known nor guaranteed [114]. There are clear arguments for the use of an avian vaccine. These include the protection of some domestic poultry species. Following the detection of WNV in geese in Israel, vaccination using both live [115] and inactivated vaccines has been effective in preventing disease [111]. 

In addition, it would be desirable to protect rare and captive species from infection. For example, vaccination has been shown to be effective in the endangered nēnē or Hawaiian goose (*Branta sandvicensis*) [112]. In the case of raptors, intra-muscular vaccination of a range of birds of prey with an inactivated whole virus vaccine was effective in stimulating anti-WNV antibodies [116]. Partial protection has been observed in large falcons vaccinated with commercial WNV vaccines [85]. Routine vaccination of birds when entering captivity would protect against risk of infection from seasonal emergence of WNV in local mosquito populations. Vaccination of wild reservoir hosts could also contribute to a decrease in the spread of WNV and might reduce spill-over infection into the human and equine populations.

Despite clear reasons to vaccinate, there are economic barriers for the development and commercialization of avian vaccines, such as the relatively small target population (captive birds of prey), particularly considering that most domestic poultry species appear to be refractory to the disease. Other factors include the difficulty in delivering vaccines to wild bird populations and the apparent poor performance of oral compared to intra-muscular immunization in birds [117]. Another potential problem is the inability to differentiate vaccinated from infected animals using standard serology tests. The widespread introduction of vaccination could negate the use of serological methods for serosurveillance studies and for the screening of domestic birds, such as racing pigeons (*Columba livia domestica*), destined for export to certain WNV-free countries. 

## 10. Conclusions

Birds of prey are clearly susceptible to natural infection with surveys both in North America and Europe reporting several clinical cases and deaths. Captive or free birds admitted to WRC are an excellent source of information of the WNV status of a region. Seropositive birds of prey are also regularly reported, suggesting exposure to virus, presumably infection, and then recovery to full health. Various owls, eagles and hawks appear to be more susceptible to clinical disease. Of those, Northern goshawks seem especially susceptible to the infection as they have been repeatedly involved in WNV outbreaks, most notably since the emergence of the WNV lineage 2 in Europe [25,63,71,81]. 

As mentioned in the introduction of this review, the terms reservoir, spreader, and sentinel, although different in definition, are tightly interrelated. The three definitions have one common aspect which is that all reservoirs, amplifiers, and sentinel species are susceptible “likely to be infected” to WNV infection. The differences between the three terms are determined by the species’ geographical area covered, their ability to infect other hosts, and the probability to detect infection by surveillance methods in a determined species. 

A common feature of experimental infection in birds of prey is the induction of mosquito-infectious viremia in 5 of the 8 tested species. In those species, viremia levels were high and lasted between 2 and 6 days [31], providing strong evidence that they can act as reservoir species and participate in the maintenance of WNV [47]. Raptors present the added risk of infection through ingestion of infected prey. However, results from experimental infections indicate that mosquito-bite infection is more efficient than oral transmission under laboratory settings [29,31,49]. The role of raptors as WNV reservoirs has also been highlighted by the preference of *Cx. pipiens* mosquitoes to feed on them over other bird species, increasing their likelihood of infection and/or transmitting the virus to naïve mosquitoes [52,54]. 

Identifying target species is important in designing cost-effective surveillance for monitoring both WNV seasonal emergence and introduction to new areas. Corvid species have been historically one of the targeted sentinel species based on mass mortality events observed in North America. However, mass mortalities have not been a feature of WNV infection in Europe. [118]. In this sense, targeting specific raptor species as disease sentinels may be beneficial. Different studies carried out in both Europe and North America indicate that raptors are among the birds more frequently infected during WNV outbreaks [46,69]. Some raptors species will more often present with clinical signs and lesions making them easier to detect, while less sensitive raptors or avian species might not be tested. Considering the data for Great horned owls, Red-tailed hawks and Sharp-shinned hawks have been regularly reported for WNV from carcasses or birds in WRC in North America. Similarly, Northern goshawks are often reported as WNV positive in Europe despite their smaller population in comparison to other raptors. 

Interestingly, most of the more commonly detected raptors are also more susceptible to develop neurological signs and clinical disease, such as Northern goshawks, Great horned owl, or Sharp-shinned hawk. On the other hand, the bias in disease reporting in some species, for example, certain species of owls and hawks, and certainly Northern goshawks, may result from human efforts to detect the virus in this species because of previous reports. 

One factor that may cause bias in WNV detection amongst raptors is their population size. However, WNV cases are frequently reported in Northern goshawks and Great grey owls in Europe and US, respectively, despite their small population size in comparison to other raptors, suggesting increased susceptibility to disease in these species. In this sense, it is important to remember that several susceptible raptors present with non-specific clinical signs, ophthalmologic and trauma-related lesions, or other concomitant infectious diseases to WRC [34,45,67,68]. Therefore, WNV should be considered as a differential diagnostic in most raptors admitted at WRC. This is important in species admitted to WRC for other lesions which may be important sentinels and more cost-effective than active surveillance in healthy wild raptors. 

Raptors sit at the top of the food chain and some species feed mainly on WNV-susceptible small birds, therefore increasing the opportunity for infection. Interestingly, some of the most commonly WNV positive detected raptors species, both in North America and Europe, such as the Northern goshawks, whose diet includes high numbers of susceptible WNV birds as corvids and small birds [62,119,120,121]. While focusing surveillance on raptors should increase the potential for detection of WNV in an area, their role as sentinels is questionable because they are likely to be infectious, and lack of infectiousness is a key criteria for ideal sentinel species [29].

Another factor that stresses the importance of birds of prey for surveillance is the geographical area they cover [29]. In past decades, the impact of urbanization with loss of indigenous bird species, the spread of alien species, and habitat loss and fragmentation have increased the movement of raptors. These have affected raptor populations with some species using sub-urban areas. Raptors have proven their ability to thrive in urban areas, such as the Red-tailed hawk [122], Sharp-shinned Hawk [123], as well as more recent urban colonists, such as the Northern goshawk [120], emphasizing their suitable role as sentinels. 

The role of raptors as spreaders is more contentious than their role as WNV reservoirs and sentinels. The migratory behavior of raptors may provide a significant contribution to long- (intercontinental) and short- (neighboring countries) distance movement of the virus. Recently, a study was able to demonstrate the relationship between WNV L1 circulation in the US and the flight pathways of terrestrial birds [124]. Moreover, during the WNV epidemic that occurred in Europe in 2018, phylogenetic analysis identified identical viruses from Spain and Austria, and viral strains isolated in Germany were reported to be descended from WNV strains isolated in the Czech Republic, suggesting that migratory birds may have been involved in the spatial spread of the virus. Nonetheless, their role and which species may have been involved in these translocations are not clear [39]. Furthermore, raptor populations are in general considerably smaller than other susceptible WNV bird species, such as those belonging to Corvidae and Passeridae families, and raptors tend not to congregate also decreasing their role as spreaders through contact and vector transmission. Therefore, their role as spreaders is likely to be less important in comparison to other avian species.

Overall, sustained wild bird surveillance programs are crucial for the early detection of WNV and other zoonotic viruses. Including raptor species as target surveillance species for WNV detection in WRC, as well as in the wild, may provide several advantages given their regular access to potentially infected raptors. Early detection of the pathogen will allow the establishment of effective measures to prevent or mitigate the effect of WNV on human populations, as well as to protect other susceptible endangered species. 

## Figures and Tables

**Figure 1 vaccines-08-00550-f001:**
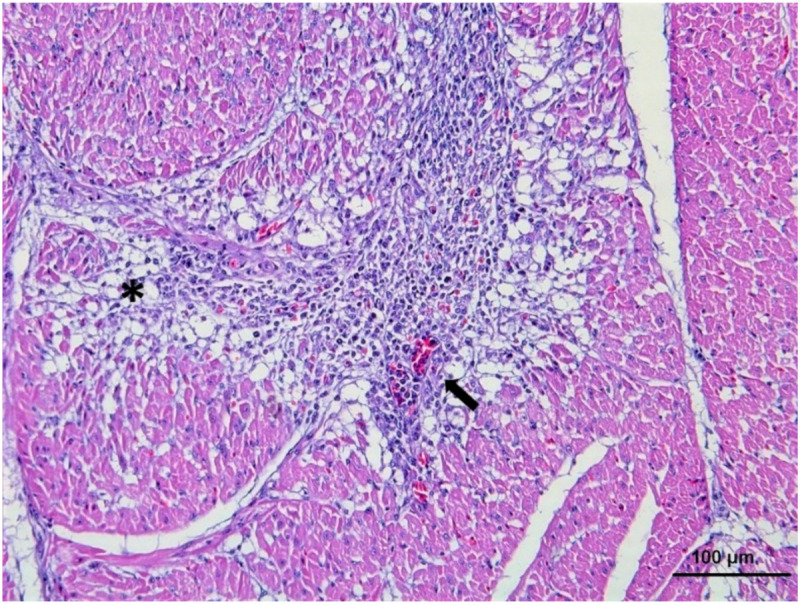
Lymphoplasmacytic myocarditis in the heart of a Gyrfalcon experimentally infected with L1 WNV. Lesions are characterized by myocardial necrosis and degeneration with lymphocytes and plasma cells (asterisk) and lymphoplasmacytic perivascular cuffs (arrow). Hematoxylin and eosin stain (HE).

**Figure 2 vaccines-08-00550-f002:**
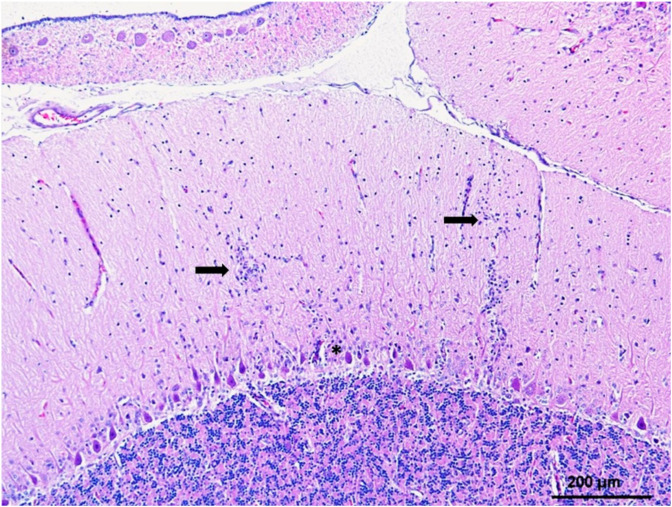
Mild lymphoplasmacytic encephalitis affecting the cerebellum of an experimentally infected Gyrfalcon with WNV L1. The lesion is characterized by few multifocal small glial nodules and lymphoplasmacytic perivascular cuffs in the molecular layer (arrows) and scarce numbers of lymphocytes and plasma cells around Purkinje neurons (asterisk). HE stain.

**Table 1 vaccines-08-00550-t001:** Clinical signs, gross and histopathological lesions, and WNV antigen detection via immunohistochemistry (IHC) in birds of prey infected with WNV.

Raptor Species	Clinical Signs			Gross Lesions ^d^	Histological Lesions		IHC	References
	Neurologic ^a^	Ocular ^b^	Non-Specific ^c^		CNS	Others		
Bald eagle (*Haliaeetus leucocephalus*)	HT, TR, AC, HLR, MD, IN	NY, BL	EM, FA, RE, DP	Brain, eyes, heart,	Brain, spinal cord	Heart, liver, kidney	Brain, heart, spleen, kidney, endocrine, lung, eye, endothelium	[17,18,45,66,74,75,76]
Cooper’s hawk (*Accipiter cooperii*)	HT, TR, AC, MD, HLR, DY, SE, DE	NY, BL	EM, FA	Cachexia, spleen	Brain, spinal cord	Eye, heart, pancreas, pectoral muscle, liver, kidney	Kidney, heart, brain, spinal cord, eye, pancreas, lung, spleen, GIT	[17,18,45,73,77,78]
Northern goshawk (*Accipiter gentilis*)	TR, IN, UN, SE, AC, HLR, HT	BL	DH, EM	Liver, spleen, heart, brain	Brain, spinal cord	Eye, heart, liver, pectoral muscle, thyroid, pancreas, spleen, eye	Brain, spinal cord, spleen, liver, eye, heart, kidney, lung, muscle	[17,28,63,70,71,73,79,80,81,82]
Red-tailed hawk (*Buteo jamaicensis*)	AC, HT, TR, HLR, CO, SE, DY	APu, NY	EM, DH, RE, HY, GI, FA,	Heart, malacia	Brain, spinal cord	Heart, liver, kidney, pancreas, pectoral muscle, arteries	Brain, eye, Kidney, heart, brain, spinal cord, GIT, liver, lung, spleen, pectoral muscle	[17,18,31,45,73,77,78,83]
Sharped-shinned hawk (*Accipiter striatus*)	AC, HT, AP, DY	APu	Abs	ND	Brain	Heart	Heart, liver, lung, kidney, spleen, GIT, brain, pectoral muscle	[18,45,73]
Peregrine falcon (*Falco peregrinus*)	Abs	Abs	Abs	ND	Brain	ND	ND	[18,45,76]
American kestrel (*Falco sparverius*)	AC, MD, HT, DY,	APu	EM, DH. FA,	Heart	Brain	Arteries, heart, liver, kidney	ND	[18,31]
Great horned owl (*Bubo virginianus*)	HT, TR, AC, HLR, MD, SE, CO, IN, DY	APu, NY	RE, EM, FA, GI, HY, DP	PA, BH, SA, GIT, cachexia, heart, malacia	Brain, spinal cord	Heart, liver, kidney, spleen, arteries, eye, pancreas	Brain, heart, kidney, liver, spleen, GIT, endocrine	[17,18,31,45,72,73,74,80]
Eastern screech owl (*Megascops asio*)	Abs/ SE	ND	FA, RE, EM, DP	PA, heart	Brain	Heart, pectoral muscle, kidney, phlebitis	Heart, lung, kidney, spleen, GIT	[18,49,73]
Great grey owl (*Strix nebulosa*)	Abs	ND	Abs/FA, EM	Liver, spleen, heart, BH, eye	Brain, spinal cord, sciatic nerve	Liver, spleen, kidney, lung, heart, GIT	brain, heart, kidney, liver, lungs, spleen, GIT, endothelium, spinal cord, endocrine, gonad, sciatic nerve, eye, pectoral muscle, thymus	[67,71,74]
Barred owl (*Strix varia*)	TR, IN, HT, SE	Abs	EM, RE, DP	Cachexia, Absent	Brain	Heart, liver, spleen, kidney, eye	Heart, kidney, spleen, lung, brain, liver, GIT	[45,67,72,73]
Swainson’s hawk (*Buteo swainsoni*)	Abs	ND	EM, FA, GI, HY, DH, DP	ND	ND	ND	ND	[18]
Barn owl (*Tyto alba*)	Abs/ AC, HLR, MD	ND	EM, DH, FA, DP	ND	ND	Liver	ND	[18,31]
Ferruginous hawk (*Buteo regalis*)	HT, TR, MD, AC, HLR, MD	NY, APu	EM, CP, DV, FA, GI, HY, DP	ND	Brain	Heart, pectoral muscle	Heart	[18,84]
Golden eagle (*Aquila chrysaetos*)	AC, MD, HLR, TR, CO, SE, DYS	APu, NY	DH, RE, EM, FA, GI, DP	Absent	Brain	Liver, eye, heart, kidney	Brain, eye, heart, kidney	[18,19,25,31,75]
Gyr falcon (*Falco rusticolus*) and hybrid falcons	HT, AC, SE	ND	CP, FA, EM, DH, EC, RE	Spleen, heart	Brain	Heart, arteries	brain, heart, liver, kidney, pancreas, peripheral nerves, skeletal muscle	[25,33,79,85]
Sparrow hawk (*Accipiter Nisus*)	IN, TR, SE, AC, HT	BL	EM, DP	ND	ND	ND	ND	[25]
Harris’s hawk (*Parabuteo unicinctus*)	IN, TR, SE, AC, HT	BL	EM, DP	ND	ND	ND	ND	[25]
Snowy owl (*Bubo scandiacus*)	IN, HT, TR, SE	ND	RE, DP	Spleen	Brain, spinal cord, sciatic nerve	Heart, liver, kidney, spleen, GIT, pectoral muscle	Brain, heart, liver, kidney, intestine, pancreas, lung, gonad, spinal cord, GIT, spleen, skin, trachea, endocrine, pectoral muscle, thymus	[30,66,72,74]
Short-eared owl (*Asio flammeus*)	UN	ND	ND	ND	ND	heart	Heart, liver, kidney, lung	[72]
Boreal owl (*Aegolius funereus*)	ND	ND	ND	UN	Brain	Liver, spleen, kidney, heart, pectoral muscle	Heart, liver, lung, kidney, spleen, GIT, eye, pectoral muscle, thymus, gonad, endocrine, trachea, pancreas, spinal cord	[30,73,74]
Northern-hawk owl (*Surnia ulula*)	ND	ND	ND	UN	Brain, spinal cord, sciatic nerve	Liver, spleen, kidney, heart, pectoral muscle	Brain, spinal cord, liver, kidney, heart, pancreas, skin, trachea, GIT, endocrine, gonad, eye, thymus	[30,74]
Saw-whet owl (*Aegolius acadicus*)	ND	ND	ND	UN	Brain, spinal cord, sciatic nerve	Liver, spleen, kidney, heart, pectoral muscle	Brain, liver, spleen, kidney, lung, heart, pancreas, GIT, skin, endocrine, gonad, sciatic nerve, thymus	[30,74]
Bearded vulture (*Gypaetus barbatus*)	Abs/UN	ND	Abs	ND	ND	ND	ND	[63]
Spanish imperial eagle (*Aquila adalberti*)	ND	ND	FA	ND	Brain	Liver, intestine, heart	Brain, GIT, liver, kidney, lung	[69]
Bonelli’s eagle (*Aquila fasciata*)	IN	BL	DP	ND	ND	ND	ND	[19]
Prairie falcon (*Falco mexicanus*)	Abs	Abs	FA	ND	Brain	ND	ND	[18,45]
Red shouldered hawk (*Buteo lineatus*)	ND	ND	ND	ND	ND	ND	Heart, brain, GIT	[73]
Long-eared owl (*Asio otus*) Northern harrier (*Circus cyaneus*) Rough-ledged hawk (*Buteo lagopus*)	Abs	ND	EM, FA, GI, DP	ND	ND	ND	ND	[18]

^a^ Neurological signs: head tilt and/or opisthotonos (HT), head and/or body tremors (TR), ataxia and/or circling (AC), abnormal positioning of tongue and/or tail and wings (AP), hind-limb paresis and/or paralysis and/or rigidity or monoplegia (HLR), hypersensitivity and/or increased aggression (HA), mentally dull/subdued/disorientated (MD), seizures/convulsion (SE), coma (CO), incoordination (IN), dysphagia (DY), ^b^ Ocular signs: nystagmus (NY), abnormal pupillary response (APu), blindness or visual impairment (BL). ^c^ Non-specific signs: crouched body postures/dropping wings (CP), depression/debilitation/lethargy/apathy (DP), feather abnormalities and/ or poor feather condition (FA), dehydration (DH), recumbency (RE), emaciation/low weight/inappetence (EM), decreased vocalizations/behavior change (DV), gastrointestinal abnormalities, including regurgitation and/or vomiting, dilation, and/or abnormal excrements (GIT), hyperthermia (HY). ^d^ Gross lesions: Pectoral muscle atrophy (PA), serous atrophy fat/bone marrow (SA), gastrointestinal lesions in proventriculus, ventriculus and intestine such as dilation and/or hemorrhages (GIT), brain hemorrhage (BA). Absent signs or lesions (Abs), unspecified but present signs or lesions (UN), not described signs or lesions (ND). Histological lesions and IHC: Gastrointestinal tract (GIT), endocrine (thyroid and /or parathyroid and/or adrenal).

**Table 2 vaccines-08-00550-t002:** Experimental infections of raptors with WNV.

Order	Family	Species	Tested Strain	Competent Host ^a^	Mortality	Oral Infection ^b^	Infection by Direct Contact ^c^	Viral Shedding		Viral Detection in Organs	Viral Detection in Feathers	Reference
								Oral	Cloacal			
Falconiformes	*Falconidae*	*Falco sparverius*	NY99 (L1)	YES	NO	YES(only in 1/8)	NO	YES	YES	YES (14 dpi)	NA	[29,31]
		*Falco rusticolus*	NY99 (L1)	YES	33%	NA	NA	YES	YES	YES (14 dpi)	NA	[33]
			AUS09 (L2)	YES	33%	NA	NA	YES	YES	YES (21 dpi)	NA	
		*Falco tinunculus*	Ar-248 (L1)	NO	NO	NA	NA	NA	NA	NA	NA	[89]
		*F. rusticolus x F. cherrug*	NY99 (L1)	NO	NO	NA	NA	YES	YES	YES (14 dpi)	YES	[32]
Accipitriformes	*Accipitridae*	*Buteo jamaicensis*	NY99 (L1)	YES	NO	YES	NO	YES	YES	YES (up to 27 dpi)	NA	[31]
Strigiformes	*Strigidae*	*Bubo virginianus*	NY99 (L1)	YES	NO	YES	NA	YES	YES	NO or at very low levels (14 dpi)	NA	[29,31]
		*Megascops asio*	NY99 (L1)	YES	40%	YES (only in 1/5)	NO	YES	YES	YES (14 dpi)	YES	[49]
	*Tytonidae*	*Tyto alba*	NY99 (L1)	NO	NO	NA	NA	YES	NO	NO (15 dpi)	NA	[31]

^a^ A species is considered a competent host for WNV when it develops a viremia ≥ 10^5^ PFU/mL [48]. ^b^ through ingestion of infected mice. ^c^ Direct contact with experimentally infected conspecifics. NA: not assayed. dpi: days post-inoculation.

**Table 3 vaccines-08-00550-t003:** Details of the main epidemiological studies carried out in North America in which raptor species were found infected with WNV.

Order	Family	Species	Scientific Name	Captive vs. Wild	Positive/Tested ^a^	Detection ^b^	Location	Year	Reference
Accipitriformes	Accipitridae	Cooper’s hawk	*Accipiter cooperii*	Wild	10/103	IHC/virus isolation	US (Georgia)	2001–2004	[73]
				Wild	3/3	ELISA + SNT	US (Colorado)	2002–2005	[18]
				Captive/Wild	3/100	RT-PCR	Canada (Ontario)	2001–2014	[34]
				Wild	10/11	IHC/RT-PCR	US (Minnesota)	2001–2014	[77]
		Northern goshawk	*Accipiter gentilis*	Wild	1/1	RT-PCR/virus isolation	US (Various)	2002	[45]
				Captive/Wild	3/23	RT-PCR	Canada (Ontario)	2001–2014	[34]
				9 Captive/3 Wild	12/12	IHC/ RT-PCR	US (Minnesota)	2002–2003	[80]
		Sharp-shinned hawk	*Accipiter striatus*	Wild	8/40	IHC/virus isolation	US (Georgia)	2001–2004	[73]
				Wild	1/1	RT-PCR	US (Virginia)	2003	[68]
				Wild	2/5	ELISA + SNT	US (Colorado)	2002–2005	[18]
				Wild	2/5	RT-PCR/virus isolation	US (Various)	2002	[45]
				Captive/Wild	10/91	PCR	Canada (Ontario)	2001–2014	[34]
		Golden eagle	*Aquila chrysaetos*	Wild	4/12	ELISA + SNT	US (Colorado)	2002–2005	[18]
				Captive/Wild	3	IHC/RT-PCR	US (Minnesota)	2004–2013	[75]
		Red-tailed hawk	*Buteo jamaicensis*	Wild	10/56	IHC/virus isolation	US (Georgia)	2001–2004	[73]
				Wild	1	Virus isolation	US (New York)	2000	[61]
				Wild	15/20	RT-PCR	US (Virginia)	2003	[68]
				Wild	18/33	ELISA + SNT	US (Colorado)	2002–2005	[18]
				Wild	10/20	RT-PCR/virus isolation	US (Various)	2002	[45]
				Captive/Wild	21/300	RT-PCR	Canada (Ontario)	2001–2014	[34]
				Wild	11/13	IHC/RT-PCR	US (Minnesota)	2002–2003	[77]
		Roughhawk-legged	*Buteo lagopus*	Captive/Wild	1/20	RT-PCR	Canada (Ontario)	2001–2014	[34]
		Red-shouldered hawk	*Buteo lineatus*	Wild	2/43	IHC/virus isolation	US (Georgia)	2001–2004	[73]
		Broad wing hawk	*Buteo platypterus*	Wild	2/2	RT-PCR	US (Virginia)	2003	[68]
		Ferruginous hawk	*Buteo regalis*	Wild	2/2	ELISA + SNT	US (Colorado)	2002–2005	[18]
		Swainson’s hawk	*Buteo swainsoni*	Wild	32/56	ELISA + SNT	US (Colorado)	2002–2005	[18]
		Bald eagle	*Haliaeetus leucocephalus*	Captive/Wild	8/95	RT-PCR	US (Virginia)	Up to 2003	[76]
				Wild	9/9	Virus isolation	US (Utah)	2013	[20]
				Wild	2/2	RT-PCR	US (Virginia)	2003	[68]
				Wild	2/3	RT-PCR/virus isolation	US (Various)	2002	[45]
				Captive/Wild	4/80	RT-PCR	Canada (Ontario)	2001–2014	[34]
				Captive/Wild	15	IHC/RT-PCR	US (Minnesota)	2004–2013	[75]
		Mississippi kite	*Ictinia mississippiensis*	Wild	1/1	ELISA + SNT	US (Colorado)	2002–2005	[18]
	Pandionidae	Osprey	*Pandion haliaetus*	Wild	1/5	IHC/virus isolation	US (Georgia)	2001–2004	[73]
Cathartiformes	Cathartidae	Black vulture	*Coragyps atratus*	Wild	1/1	RT-PCR	US (Virginia)	2003	[68]
Falconiformes	Falconidae	Merlin	*Falco columbarius*	Captive/Wild	5/50	RT-PCR	Canada (Ontario)	2001–2014	[34]
		Prairie falcon	*Falco mexicanus*	Wild	1/1	RT-PCR/virus isolation	US (Various)	2002	[45]
		Peregrine falcon	*Falco peregrinus*	Captive/Wild	3/16	RT-PCR	US (Virginia)	Up to 2004	[76]
				Wild	1/2	RT-PCR	US (Virginia)	2003	[68]
				Wild	1/2	ELISA + SNT	US (Colorado)	2002–2005	[18]
		American kestrel	*Falco sparverius*	Wild	1/2	RT-PCR	US (Virginia)	2003	[68]
				Wild	24/68	ELISA + SNT	US (Colorado)	2002–2005	[18]
				Captive/Wild	5/71	RT-PCR	Canada (Ontario)	2001–2021	[34]
Strigiformes	Strigidae	Northern saw-whet owl	*Aegolius acadicus*	Captive	12/12	RT-PCR	Canada (Ontario)	2002	[30]
		Boreal owl	*Aegolius funereus*	Captive	10/11	RT-PCR	Canada (Ontario)	2002	[30]
		Short-eared owl	*Asio flammeus*	Captive/Wild	1	IHC/ RT-PCR	US (Michigan/Ohio)	2002	[72]
				Captive	2/2	RT-PCR	Canada (Ontario)	2002	[30]
		Long-eared owl	*Asio otus*	Captive	3/3	RT-PCR	Canada (Ontario)	2002	[30]
		Snowy owl	*Bubo scandiacus*	Captive/Wild	7	IHC/ RT-PCR	US (Michigan/Ohio)	2002	[72]
				Captive	11/11	RT-PCR	Canada (Ontario)	2002	[30]
		Great horned owl	*Bubo virginianus*	Wild	1/18	IHC/virus isolation	US (Georgia)	2001–2004	[73]
				Captive/Wild	4	IHC/ RT-PCR	US (Michigan/Ohio)	2002	[72]
				Captive	1/1	RT-PCR	Canada (Ontario)	2002	[30]
				Wild	16/22	RT-PCR	US (Virginia)	2003	[68]
				Wild	33/56	ELISA + SNT	US (Colorado)	2002–2005	[18]
				Wild	9/16	RT-PCR/virus isolation	US (Various)	2002	[45]
				Captive/Wild	18/225	RT-PCR	Canada (Ontario)	2001–2014	[34]
				1 Captive/24 Wild	12/25	IHC/RT-PCR	US (Minnesota)	2002–2003	[80]
		Northern pygmy owl	*Glaucidium californicum*	Captive	1/1	RT-PCR	Canada (Ontario)	2002	[30]
		Eastern Screech Owl	*Megascops asio*	Wild	4/42	IHC/virus isolation	US (Georgia)	2001–2004	[73]
				Wild	4/6	ELISA + SNT	US (Colorado)	2002–2005	[18]
				Captive/Wild	1/100	RT-PCR	Canada (Ontario)	2001–2014	[34]
		Flammulated owl	*Otus flammeolus*	Captive	1/1	RT-PCR	Canada (Ontario)	2002	[30]
		Tawny owl	*Strix aluco*	Captive	1/1	RT-PCR	Canada (Ontario)	2002	[30]
		Great gray owl	*Strix nebulosa*	Captive	21/23	RT-PCR	Canada (Ontario)	2002	[30]
				Wild	6	Antigen detection	US (Minnesota)	2003–2005	[67]
				Captive/Wild	4/27	RT-PCR	Canada (Ontario)	2001–2014	[34]
		Spotted owl	*Strix occidentalis*	Captive	1/1	RT-PCR	Canada (Ontario)	2002	[30]
		Barred owl	*Strix varia*	Wild	4/27	IHC/virus isolation	US (Georgia)	2001–2004	[73]
				Captive/Wild	1	IHC/RT-PCR	US (Michigan/Ohio)	2002	[72]
				Wild	2	Antigen detection	US (Minnesota)	2003–2005	[67]
				Captive/Wild	1/50	RT-PCR	Canada (Ontario)	2001–2014	[34]
		Northern hawk owl	*Surnia ulula*	Captive	17/17	RT-PCR	Canada (Ontario)	2002	[30]
	Tytonidae	Barn owl	*Tyto alba*	Wild	1/3	RT-PCR	US (Virginia)	2003	[68]

^a^ Positive/Tested: When number of tested was not available, the number indicates positives only. ^b^ Detection: IHC: Immunohistochemistry; SNT: Serum Neutralisation Test.

**Table 4 vaccines-08-00550-t004:** Details of the main epidemiological studies carried out in Europe in which raptor species were found infected with WNV.

Order	Family	Species	Scientific Name	Captive vs. Wild	Positive/Tested ^a^	Detection ^b^	L ^c^	Country	Year	Reference
Accipitriformes	Accipitridae	Northern goshawk	*Accipiter gentilis*	Wild	4	RT-PCR	2	Hungary	2004–2005	[97]
				Wild	35/57	RT-PCR	2	Hungary/ Austria	2008–2009	[46]
				2 Captive + 1 Wild	3	2 RT-PCR 1ELISA	2	Spain	2017	[63]
				Captive + Wild	3	ELISA + SNT		Slovakia	2012–2014	[98]
				Captive + Wild	5	RT-PCR	2	Hungary	2004	[70]
				Captive	2	RT-PCR	2	Czech Republic	2017	[81]
				12 Captive + 5 Wild	17	RT-PCR	2	Czech Republic	2018	[25]
				Captive * + Wild	1	ELISA + SNT		Germany	2018	[39]
				Captive + Wild	3/3	RT-PCR	2	Serbia	2012	[99]
				Wild	1	RT-PCR	2	Italy	2012	[82]
				Wild	5/5	RT-PCR	2	Austria	2008	[79]
				1 Captive + 1 Wild	2	RT-PCR	2	Germany	2018	[71]
				Wild	10	RT-PCR + ELISA + IHC		Germany	2019	[28]
		Sparrowhawk	*Accipiter nisus*	Wild	2	RT-PCR	2	Hungary	2005	[97]
				Wild	1/12	RT-PCR	2	Hungary	2009	[46]
		Eurasian sparrowhawk	*Accipiter nisus*	Captive	1	RT-PCR	2	Hungary	2004	[70]
				Wild	2	RT-PCR	2	Czech Republic	2018	[81]
		Spanish imperial eagle	*Aquila adalberti*	Wild	1	ELISA + SNT		Spain	2009	[100]
				4 Captive + 4 Wild	8	RT-PCR		Spain	2001–2005	[69]
		Golden eagle	*Aquila chrysaetos*	Captive + Wild	1	ELISA + SNT		Slovakia	2012-2014	[98]
				Captive	1	RT-PCR	2	Czech Republic	2018	[81]
				1 Captive + 1 Wild	2	RT-PCR	1	Spain	2007	[19]
		Bonelli’s eagle	*Aquila fasciata*	Captive	1	RT-PCR	1	Spain	2007	[19]
		Booted eagle	*Aquila pennata*	Captive	1/5	ELISA + SNT		Spain	2011–2014	[101]
		Common buzzard	*Buteo buteo*	Captive + Wild	1	ELISA + SNT		Slovakia	2012–2014	[98]
			*Buteo buteo*	Captive * + Wild	7	ELISA + SNT		Germany	2017–2018	[39]
		Short-toed snake eagle	*Circaetus gallicus*	Wild	5/9	ELISA + SNT		Spain	2010–2011	[102]
			*Circaetus gallicus*	Captive	3/5	ELISA + SNT		Spain	2011–2014	[101]
		Western marsh-harrier	*Circus aeruginosus*	Captive + Wild	1	ELISA + SNT		Slovakia	2012–2014	[98]
				Captive * + Wild	2	ELISA + SNT		Germany	2018	[39]
		Hen Harrier	*Circus cyaneus*	Wild	1/1	ELISA + SNT		Spain	2010–2014	[102]
		Montagu’s harrier	*Circus pygargus*	Captive	1/2	ELISA + SNT		Spain	2011–2014	[101]
		Bearded vulture	*Gypaetus barbatus*	Wild	1/1	RT-PCR	2	Austria	2008	[46]
				Captive	13	ELISA + SNT		Spain	2017	[63]
		White-tailed Eagle	*Haliaeetus albicilla*	Captive * + Wild	1	ELISA + SNT		Germany	2018	[39]
				Captive + Wild	1/8	ELISA + SNT		Serbia	2012	[99]
		Black Kite	*Milvus migrans*	Wild	1/4	ELISA + SNT		Spain	2010–2012	[102]
				Captive * + Wild	1	ELISA + SNT		Germany	2018	[39]
		Red Kite	*Milvus milvus*	Wild	1/7	ELISA + SNT		Spain	2010–2013	[102]
		Harris hawk	*Parabuteo unicinctus*	Captive	2/2	RT-PCR	2	Hungary	2008	[46]
				Captive	1	RT-PCR	2	Czech Republic	2018	[81]
	Pandionidae	Osprey	*Pandion haliaetus*	Captive + Wild	1	ELISA + SNT		Slovakia	2012–2014	[98]
Falconiformes	Falconidae	Peregrine falcon	*Falco peregrinus*	Wild	1/1	RT-PCR	2	Austria	2009	[46]
		Gyrfalcon	*Falco rusticolus*	Wild	3/4	RT-PCR	2	Hungary	2008	[46]
				Wild	1/1	RT-PCR	2	Austria	2008	[79]
		European Kestrel	*Falco tinnunculus*	Captive * + Wild	3	ELISA + SNT		Germany	2017–2018	[39]
Strigiformes	Strigidae	Long-eared owl	*Asio otus*	Captive	2/10	ELISA + SNT		Spain	2011–2014	[101]
		Little owl	*Athene noctua*	Wild	1	RT-PCR	1	Italy	2011	[103]
		Eurasian eagle-owl	*Bubo bubo*	Captive + Wild	1	ELISA + SNT		Slovakia	2012–2014	[98]
				Captive	1/24	ELISA + SNT		Spain	2011–2014	[101]
		Snowy owl	*Nyctea scandiaca*	Wild	1/1	RT-PCR	2	Austria	2009	[46]
		Scops owl	*Otus scops*	Wild	2/6	ELISA + SNT		France	2005–2006	[104]
		European scops owl	*Otus scops*	Captive	1/5	ELISA + SNT		Spain	2011–2014	[101]
		Great grey owl	*Strix nebulosa*	Captive	2	RT-PCR	2	Germany	2018	[71]

^a^ Positive/Tested: When the number of tested was not available, the number indicates positives only. ^b^ Detection: IHC: Immunohistochemistry; SNT: Serum Neutralisation Test; ^c^ L: Lineage. *: Zoo birds excluded.

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
