# Peer review of "The Role of Birds of Prey in West Nile Virus Epidemiology"

_vaccines, 2020, doi:10.3390/vaccines8030550_

Round 1
Reviewer 1 Report
This manuscript is a comprehensive review of birds of prey in West Nile virus ecology. The manuscript is well-written with really my only comments would be differences writing style. I have no major concerns and recommend acceptance.
Author Response
The author's are thankful for the reviewer's comment. The language and style of the manuscript have now been revised.
Reviewer 2 Report
This is very comprehensive and well written review summarizing the role of birds of prey in epidemiology of WNV and their potential for WNV surveillance. The authors summarized data on WNV in raptors from Europe and North America with emphasis on virus infection and immunity, clinical manifestations, gross pathology and histopathology in various raptor species. In addition, they summarized data on experimental WNV infections and mechanisms of WNV transmission to the birds of prey. Separate section of review attributed to the diagnosis and surveillance of WNV in birds of prey, suggesting its potential important contribution to WNV epidemiology. Finally, the authors summarized situation with WNV vaccine for avian species and provide some data on experimental vaccines.
The manuscript is well written and all data are well described and supported by multiple references and comprehensive summary tables. Overall this review provides a nice summary of different aspects of WNV infection in birds of prey and will be of great interest to research community, especially for the researches working on WNV. Corrections and suggestions to the authors are minimal and presented in the comments to the authors.
This is a nice and comprehensive review of the role of birds of prey WNV spread and surveillance. Only a few minor comments:
P12. Table 2. There is a superscript c in the footnote, but I do not see it in the Table.
Line 522 in Discussion section: “there are no equivalent licensed vaccines for use in birds” – not clear wording. Does it mean that there are no licensed vaccines for birds; or it means that there are some licensed vaccines for birds, but they are not as good as equine vaccine? Please clarify.
Author Response
The authors are thankful for the reviewer’s assessment. All minor comments suggested by the reviewer have been checked and amended.
P12. Table 2. There is a superscript c in the footnote, but I do not see it in the Table. (corrected line 434, table 2)
Line 522 in Discussion section: “there are no equivalent licensed vaccines for use in birds” – not clear wording. Does it mean that there are no licensed vaccines for birds; or it means that there are some licensed vaccines for birds, but they are not as good as equine vaccine? Please clarify.
-There are currently no licensed WNV vaccines for birds. (corrected line 564)
Reviewer 3 Report
1) Despite the interesting question posed by the title of this manuscript, the review is presented in a conventional format. The article would be of greater interest if there was a clearly articulated evaluation of the data for or against reservoir/spreader/sentinel status, including definitions of reservoir, spreader, and sentinel.
2) The Introduction should provide a brief description of WNV – Flaviviridae, phylogeny (lineages), no vaccine available for human use.
3) Why is WNV transmission section 6 of the manuscript (after experimental infection)? Transmission is the method birds of prey become infected and arguably should be at the beginning of the article.
4) Is there any correlation between mode of transmission and clinical outcome?
Line 54: falcon-like
Lines 56-61: move to transmission section and cite primary data rather than reviews [eg. 15].
Line 63-64: grammar needs attention. Suggest: When wildlife becomes infected, determining the clinical response and outcomes are difficult to establish when such events are seldom observed.
Line 84-85: this sentence is a non sequitur in the context of the paragraph unless qualified by referring to wild birds.
Lines 92-106: were comparable methods of infection (route, dose, virus type) used in the different studies?
Line 113: IgG and IgM are mammalian immunoglobulins
Lines 186-192: references?
Line 205: Cite references to support correlation between WNV type and pathogenicity.
Fig. 1 label figure to indicate observations made in the legend
Table 1, 2, 3 and 4. How does the information shown in these tables relate to the question posed in the article’s title?
Lines 251-254: higher than what?
Fig. 2 label figure to support observations in legend
Line 278: “associated or not” means ?
Lines 301-304: mode of infection?
Line 307: “Once the animals are infected…” – is mode of infection significant?
Line 309: mode of infection?
Lines 313-318: were experimental conditions comparable in these studies?
Line 335: cite reference
Line 346-347: provide a clearer explanation of mosquito-borne transmission. Just because a mosquito feeds on a viraemic animal does not mean that it will transmit the infection.
Lines 366-373: only one reference?
Line 410: “species will be detected”? “the infection will be detected”?
Line 424: “Conversely..”?
Line 436: “differences between their population sizes”
Line 448-450: to be comparable, numbers need to be qualified by population sizes
Line 465-512: Is there a triage (decision) tree published for diagnosing WNV infections in birds? Could one be constructed to define reservoir, spreader or sentinel?
Line 562: certain types…certain species?
Line 571: “..these have regular access to these species”?
Line 576: “preference of Culex pipiens mosquitoes”
Line 584: “..more contentious” than what?
Lines 596-597: “In that sense…In this sense..” ?
Author Response
Answers to reviewer’s comments
The authors are thankful for the reviewer’s assessment.
Reviewer Comments to Author:
1) Despite the interesting question posed by the title of this manuscript, the review is presented in a conventional format. The article would be of greater interest if there was a clearly articulated evaluation of the data for or against reservoir/spreader/sentinel status, including definitions of reservoir, spreader, and sentinel.
The review has been presented in a conventional way describing different biological aspects of WNV infection in birds of Prey. In order to better reflect the content of the review a new title has been proposed “The role of birds of prey in West Nile epidemiology”. In addition, a brief description of the requisites that characterise each term has been included in the introduction (lines 75-79) and the conclusion section has been modified and now includes a more extensive and detailed discussion evaluating the data against the reservoir/spreader/sentinel possible role of birds of prey (lines 605-678).
2) The Introduction should provide a brief description of WNV – Flaviviridae, phylogeny (lineages), no vaccine available for human use.
As requested by the reviewer the introduction has now included a brief description of WNV phylogeny and human vaccination status. (lines 38-42)
3) Why is WNV transmission section 6 of the manuscript (after experimental infection)? Transmission is the method birds of prey become infected and arguably should be at the beginning of the article.
Transmission section has now been moved to the beginning of the article. (lines 154-219)
4) Is there any correlation between mode of transmission and clinical outcome?
The only data available in this respect is extrapolated from experimental infections comparing different routes of infection in a few raptor species and using a very small number of birds per group. Therefore, it is not possible to draw conclusions in this aspect. A paragraph reviewing the results of these studies in relation to the clinical outcome has been added to the manuscript. (lines 414-422)
5) Table 1, 2, 3 and 4. How does the information shown in these tables relate to the question posed in the article’s title?
The tables gathered the information available from natural and experimental infection in birds of prey. The authors believe that the tables provide important information regarding the epidemiological role of raptors in WNV infection.
6) Line 465-512: Is there a triage (decision) tree published for diagnosing WNV infections in birds? Could one be constructed to define reservoir, spreader or sentinel?
To the best of the authors’ knowledge there is no triage decision tree published for WNV diagnosis in avian species. However, the World Organisation for animal Health (OIE) indicates the different test methods available for WNV and their purpose; see table 1 on:
https://www.oie.int/fileadmin/Home/eng/Health_standards/tahm/3.01.24_WEST_NILE.pdf.
Since it is not possible to clearly differentiate the terms reservoir, amplifier and spreader, and the overlap of the definitions in the case of raptors and WNV epidemiology. It is not possible to create a decision tree for their diagnosis in birds of prey.
___________
- The following minor comments suggested by the reviewer have been checked and amended:
Line 205: Cite references to support correlation between WNV type and pathogenicity. (lines 288-290)
Line 113: IgG and IgM are mammalian (line 130)
Line 54: falcon-like. (line 59)
Lines 56-61: move to transmission section and cite primary data rather than reviews [eg. 15]. (lines 210-213)
Line 63-64: grammar needs attention. Suggest: When wildlife becomes infected, determining the clinical response and outcomes are difficult to establish when such events are seldom observed. (lines 69-70)
Line 84-85: this sentence is a non sequitur in the context of the paragraph unless qualified by referring to wild birds. (lines 99-101)
Lines 92-106: were comparable methods of infection (route, dose, virus type) used in the different studies? (lines 113-117)
Fig. 1 label figure to indicate observations made in the legend. -The figure has been labelled.
Lines 186-192: references? (lines 272-274)
Lines 251-254: higher than what? (lines 338-340)
Fig. 2 label figure to support observations in legend. - The figure has been labelled.
Line 278: “associated or not” means ? (lines 366-367)
Lines 301-304: mode of infection? (line 392)
Line 307: “Once the animals are infected…” – is mode of infection significant? (line 396)
Line 309: mode of infection? (line 396)
Lines 313-318: were experimental conditions comparable in these studies? (lines 404-405)
Line 335: cite reference (line 435)
Line 346-347: provide a clearer explanation of mosquito-borne transmission. Just because a mosquito feeds on a viraemic animal does not mean that it will transmit the infection. (lines 161-162)
Lines 366-373: only one reference? (line 184)
Line 410: “species will be detected”? “the infection will be detected”? (line 452)
Line 424: “Conversely..”? (line 466)
Line 436: “differences between their population sizes” (line 478)
Line 448-450: to be comparable, numbers need to be qualified by population sizes (lines 488-492)
Line 562: certain types…certain species? (line 634)
Line 571: “..these have regular access to these species”? (line 675)
Line 576: “preference of Culex pipiens mosquitoes” (line 617)
Line 584: “..more contentious” than what? (line 660)
Lines 596-597: “In that sense…In this sense..” ? (line 623)
Round 2
Reviewer 3 Report
The authors have responded very constructively to the comments. The emphasis of the review is now on wild bird surveillance programmes for detecting WNV infections and vaccination is a minor part. It does beg the question - is Vaccine the most appropriate journal for this authoritative review.
Some minor points:
line 199: "..type of non-vectored transmission.." delete "type of"?
Line 400: "was" rather than "were"
Line 633-634: "..the geographical they cover.."?
Line 635: "have" rather than "has"
Line 648: "translocations"
Author Response
All minor comments suggested by the reviewer have been amended in the manuscript.